# Do birthrates contribute to sickness absence differences in women? A cohort study in Catalonia, Spain, 2012-2014

**Andrew N. March** [1]*, **Rocío Villar**[1,2,3], **Monica Ubalde-Lopez** [4], **Fernando G. Benavides**[1,2,3], **Laura Serra**[5]

**1** Center for Research in Occupational Health, Universitat Pompeu Fabra, Barcelona, Spain, **2** CIBER de Epidemiología y Salud Pública, Madrid, Spain, **3** IMIM Parc de Salut Mar, Barcelona, Spain, **4** Barcelona Institute for Global Health (ISGlobal), Barcelona, Spain, **5** Research Group on Statistics, Econometrics and Health (GRECS), University of Girona, Girona, Spain

* andrewn.march@gmail.com

**Data Availability Statement:** All relevant data are within the paper and its Supporting Information files.

## Abstract

### Aims

This study explores the differences in sickness absence trends in women according to reproductive age group and medical diagnoses.

### Methods

Data were obtained from two administrative registries: the Continuous Working Life Sample and the Catalonian Institute of Medical Evaluations from 2012 to 2014, containing 47,879 female employees. Incidence rates and incidence risk ratios derived from Poisson and negative binomial models were calculated to compare sickness absence trends among reproductive age groups based on Catalonian birthrates: early-reproductive (25–34 years old), middle-reproductive (35–44) and late-reproductive (45–54), according to diagnostic groups, selected diseases, type of contract, occupational category, and country of origin.

### Results

Younger women show a higher incidence of overall sickness absence compared to late-reproductive-aged women. Incidence risk ratios of sickness absence decreased significantly from early-reproductive to late-reproductive age for low back pain, hemorrhage in early pregnancy, nausea and vomiting, and abdominal and pelvic pain.

### Discussion

The higher incidence of sickness absence due to pregnancy-related health conditions in early-reproductive women compared to other reproductive age groups, may explain the sickness absence differences by age in women. Proper management of sickness absence related to pregnancy should be a goal to reduce the sickness absence gap between younger and older women.

**Funding:** This study was financed by the State Plan for Investigation, Development, and Innovation 2013-2016, by the Health Institute Carlos III – Subdirection General of Evaluation and Promotion of Investigation (FIS PI14/00057 – EBISA), and by the European Regional Development Fund. The authors claim no conflicts of interest. This grant was awarded to the Center for Research in Occupational Health (http://www.cisal.upf.edu/workss/eng/frontpage). The funders had no role in study design, data collection and analysis, decision to publish, or preparation of the manuscript.

**Competing interests:** The authors have declared that no competing interests exist.

## Introduction

The EU-28 average of female labor market participation for women aged 25 to 54 rose from 67% in 2001 to 72% in 2014. In Spain, the percentage of employed women in this age range increased from 53% to 62% during the same period [1]. This increase in female employment has lead to the recognition of the double burden working women often encounter as they navigate the demands of paid employment and unpaid domestic work, which may partly explain why women have higher sickness absence (SA) than men [2].

Similarly, the demands of pregnancy have been posited as another cause of higher SA in women [2]. The medical consensus is that maternity is a biological process, although health in pregnancy faces high physiological and psychological demands, which could lead to health problems [3], either work-related or not. In fact, a precipitous increase in the amount of sickness absence benefits used can be seen during pregnancy [4]. This situation may contribute to pregnancy workplace discrimination, in which pregnant women are subject to perceptions including being less committed to their job and creating more work for colleagues [5]. Even pregnancies without exposure to occupational risk factors may face symptoms such as nausea and vomiting, headaches, back pain, and fatigue [6], though even these diseases have been shown to decrease health-related quality of life in pregnant women [7]. Evidence demonstrates the relation between occupational exposures and health effects during pregnancy, such as an increase in the number of low birth weight and prematurity in relation to a heavy physical workload [8]. To deal with this situation, and protect both the health of working women and the fetus, as well as support continuity in their employment, many countries, including Spain, have developed specific social protection benefits. The most relevant social protection benefit for workers, men and women, with a health problem are sickness absence benefits. The International Labour Organization defines SA as a situation in which a worker is unable to perform the essential tasks demanded by his or her workplace as a consequence of a diversion from their habitual state of health, and it can be considered as a social right that allows the worker to be temporarily absent from the workplace due to non-permanent health-related causes while he or she receives medical care [9].

The sickness absence benefit in Spain is a social benefit from which any worker can benefit given that they have been affiliated to the Social Security System. For common disease, not occupational injury or disease, current legislation requires that the sick leave be certified by a National Health System physician in order to offer a diagnosis and recognize the absence from work as SA. SA spells are recorded for all health-related absences recognized by a National Health System physician beginning on the day the physician certifies the absence until the health condition initiating the absence is resolved, or in extreme instances, when the condition cannot be resolved and permanent disability is required. After the third day of absence, the employee's wages are covered between 60% and 70%, first by the employer, until the sixteenth day of absence when the National Social Security Institute begins to pay until the end of the SA episode [10]. Due to the differences in sickness absence benefits between countries, it is not clear whether research findings from other countries are applicable to the Spanish worker.

Compared to younger workers, a greater risk of SA in older individuals has been described [11], as well as a higher incidence in women in comparison to men, but the cause of the gender difference is poorly studied [12,13]. A study in Sweden found that female workers have higher incidence of SA during pregnancy compared to the years prior and posterior to their pregnancy [4]. Although sickness absence benefits are not designed as a family policy benefit, female workers may use it as a tool to care for their health and that of their fetus when adverse work exposures are not adequately mitigated in advance, or when family policy benefits are deficient.

In Spain, maternity leave benefits provide women 16 continuous weeks of leave paid at 100% of their normal salary. Six of these weeks must be used directly after birth, but the remaining 10 weeks may be used before or after the birth, donated to the woman's partner, or any combination thereof [14]. In 2014, of those who provided a response, 53% of women who gave birth in Catalonia described themselves as being active in the labor market [15]. In that same year, the total fertility in Spain was 1.32 [16], and a 2015 estimate suggests that the median age of the mother at first birth was 30.7 years for Spain [17]. Overall, Spain is characterized as having lower fertility and an older age at first birth than the rest of the EU [16,17].

Various Norwegian studies demonstrated that the incidence of SA during pregnancy is rising, though the cause is not apparent [18]. In fact, a study examining SA in Spain reported that obstetric diagnoses accounted for 14% of all SA episodes lasting at least 15 days in working persons without differentiating between men and women [19]. Further exploration of the behavior of SA according to reproductive age is compelling given the poorly explained gender gap in SA. Moreover, a broad description of SA across age and disease states is, at the time of this publication, insufficient to understand the potential causes of this gender gap in a Southern European state such as Spain, which has a weaker welfare state than Northern European countries, where this phenomenon is better described. The aim of this study is to compare the incidence rates of overall SA and according to specific diseases among early-, middle-, and late-reproductive age groups in female workers in Catalonia (Spain) between 2012 and 2014.

## Methods

### Study population

The study population belongs to a cohort of 64,361 working women, affiliated to the Spanish Social Security as employees (general regime), which covers all workers with formal employment. The study period went from the beginning of 2012 to the end of 2014. The sample used in this study is drawn from the Spanish WORKss cohort [20]. The WORKss cohort is the result of merging two administrative registries: the Continuous Working Life Sample (MCVL) from the Social Security Institute, and the Catalonian Institute of Medical Evaluations (ICAMS) from the Department of Health (Fig 1). A detailed description of the WORKss Cohort has previously been described by López Goméz and colleagues [20].

The MCVL, designed to help the Social Security Institute to manage contributions and payments [21], annually collects occupational information from a 4% representative sample of the Spanish population associated with the Social Security System, including SA data and employment characteristics [20]. The annual reference population for the MCVL is any individual who has contributed to or received payments from Social Security at any point during that year. Individuals are selected to the MCVL based on a simple random sample, and remain a part of the MCVL indefinitely until they contribute to or receive from the Social Security at any point over a full calendar year. New individuals enter the MCVL every year until the sample reaches 4% of the population [20]. While the MCVL is a robust sample that allows for the inclusion of individuals with infrequent employment, it excludes some categories of civil servants covered by distinct insurance funds, such as university professors and military personnel [22], which represent about 5% of the Spanish working population [20]. ICAMS records SA medical diagnoses for all resident workers in Catalonia. The record linkage between both registries was carried out following an agreement with the Social Security authority.

### Variables

The dependent variable was the total number of SA spells initiated between 2012–2014. An SA spell was defined as any absence from work due to common disease as certified by a physician,

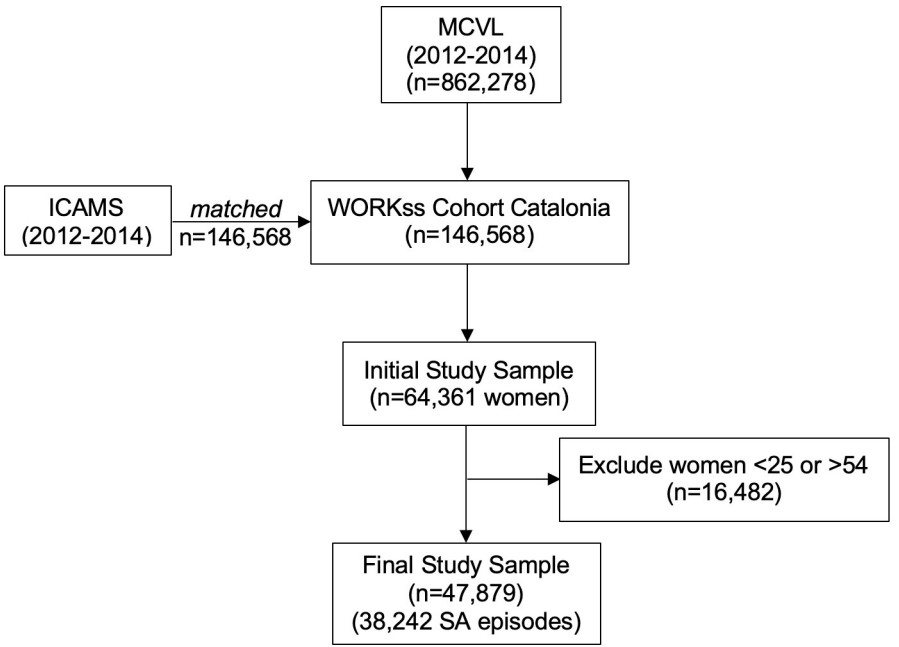

**Fig 1. Construction of WORKss cohort and final study sample.**

as is required by law. For the independent variable, women were assigned to one of three age categories based on their age as of December 31, 2012, and following birthrates from Catalonia during the study period: early-reproductive age (25–34), middle-reproductive age (35–44), and late-reproductive age (45–54). In Catalonia, between 2012 and 2014, more than 97% of the pregnancies corresponded to women with ages between 20 and 44. The birthrates in this population in the year 2014 demonstrated that women between 25 and 34 years had the highest birthrate (40.26 births/1000 persons), followed by women between 35 and 44 years (19.63 births/1000 persons). Women in the age group of 45–54 had the lowest birth rate (0.36 births/1000 persons) [15]. After excluding women under 25, because they had relatively low birth and employment rates (8.76 births/1000 persons and below 25%, respectively), and women over 54 (natality data not reported), 16,482 women were removed and the final sample size was 47,879 women, or about 4.2% of the female working population between 25 and 54 years of age in Catalonia [15].

SA medical diagnoses were coded according to the 10th edition of the International Classification of Diseases (ICD-10). Occupational category (skilled non-manual, unskilled non-manual, skilled manual, unskilled manual); type of contract (permanent, temporary); and country of origin (Spain and other EU-28 states, and all other countries) were considered as potential confounders. Occupational category and type of contract were assigned based on the situation to which the woman was affiliated for the most time over the three-year period of this study.

## Statistical analyses

SA incidence rates (IR) were calculated for women of each reproductive age group, according to diagnostic group, type of contract, occupational category, country of origin, and specific diseases. The IR was calculated as the number of SA spells per 100 person-years at risk. Person-year at risk was defined as the sum of days affiliated to the Social Security system, subtracting days on SA, and dividing by 365.33 (accounting for leap year in 2012). Confidence intervals at 95% were estimated for each of the incidence rates calculated.

Within the diagnostic groups that showed significant differences among reproductive age groups, diseases that represented at least 5% of all diagnoses in the group, were also assessed. The perinatal diagnostic group was not considered in this analysis as these codes refer exclusively to neonates.

Incidence risk ratios (IRR) were calculated using crude and adjusted Poisson and negative binomial models, with the data presented deriving from whichever of the two demonstrated the higher log likelihood. Consequently, IRR values are based on Poisson models for overall SA, and negative binomial models were used for specific diseases in order to compare the incidence rates of SA among age groups. All analyses were performed using STATA Version 13.

## Ethics

The protocol for the study described in this article (2018/8036/I), was approved by the Clinical Ethics Committee at *Parc de Salut Mar* in Barcelona, Spain. Written informed consent from those included in the cohort was not required. All data were anonymized and deidentified before the authors had access to them.

## Results

There were less women in the late-reproductive group than the early- or middle-reproductive groups. Across all age groups, women were most commonly characterized as having a permanent contract, working in an unskilled non-manual job, and originating from Spain or other EU country. The distribution of occupational and demographic characteristics were similar across all age groups (Table 1).

The 47,879 women included in the sample accumulated a total of 38,242 SA episodes from 2012 to 2014. The overall incidence rate of SA was significantly higher in early-reproductive age women (43.62 spells/100 person-years), followed by middle-reproductive age women (35.03 spells/100 person-years), and this rate was the lowest in the late-reproductive group (32.65 spells/100 person-years). This descending trend was maintained at a statistically significant level across both contract types, women with skilled non-manual and unskilled manual occupations and women born in EU-28 countries. However, there were no significant differences in the incidence rates between age groups among immigrants from outside of the EU.

**Table 1. Occupational and demographic characteristics in female workers by age group, Catalonia 2012–2014.**

|  | Early-Reproductive | | Middle-Reproductive | | Late-Reproductive | |
|---|---|---|---|---|---|---|
| **Median Age** | **31.58** | | **40.50** | | **50.50** | |
|  | **(n)** | **(%)** | **(n)** | **(%)** | **(n)** | **(%)** |
| **Contract** | | | | | | |
| Permanent | 11,153 | 64.2 | 13,163 | 73.3 | 9,888 | 78.8 |
| Temporary | 6,222 | 35.8 | 4,795 | 26.7 | 2,657 | 21.2 |
| **Occupational Category** | | | | | | |
| Skilled non-manual | 3,740 | 21.5 | 3,813 | 21.2 | 2,318 | 18.5 |
| Unskilled non-manual | 8,271 | 47.6 | 8,265 | 46.0 | 5,284 | 42.1 |
| Skilled manual | 3,131 | 18.0 | 3,127 | 17.4 | 2,312 | 18.4 |
| Unskilled manual | 2,233 | 12.9 | 2,754 | 15.3 | 2,631 | 21.0 |
| **Country of Origin** | | | | | | |
| Spain and EU-28 | 13,155 | 78.1 | 14,502 | 82.5 | 10,699 | 86.8 |
| Other | 3,668 | 21.9 | 3,077 | 17.5 | 1,628 | 13.2 |
| **Overall** | 17,375 | 100 | 17,959 | 100 | 12,545 | 100 |

Though the median duration of SA varies widely depending primarily on the diagnostic group, overall SA episodes are longer for women in the late reproductive group than for early- or middle-reproductive groups (Table 2).

In regards to diagnostic groups, as shown in Table 2, incidence rates decreased significantly between each of the three age groups, from early-reproductive to late-reproductive for infectious, obstetric, and other poorly specified diseases, and increased for neoplastic diagnoses. There were also significant differences in the incidence rates between each of the age groups for musculoskeletal diagnoses, though the trend was not linear with age. The incidence rate of the musculoskeletal group was lowest in middle-reproductive age, and highest in early-reproductive age women.

Exploring in greater detail specific diseases, gastroenteritis and colitis, low back pain, hemorrhaging during early pregnancy, and vomiting during pregnancy all demonstrated a significant decline in IR between each age group from early-reproductive women to the late-reproductive group (Table 3). The early-reproductive age group also had a significantly higher incidence rate of neck pain (IR = 0.73; 95% CI: 0.64–0.82), nausea and vomiting (IR = 0.56; 95% CI: 0.48–0.63), and abdominal and pelvic pain (IR = 0.56; 95% CI: 0.48–0.63) than the two older age groups.

In Table 4, the crude model for overall SA shows that the IRR for the early-reproductive age group (IRR = 1.09; 95% CI: 1.06–1.12) is modestly, yet significantly greater than the late-reproductive age group (IRR = 1 [reference]); adjusting for contract, occupational category, and country of origin does not change this pattern. However, when diagnostic group is also taken into account in this model for all SA episodes, this association almost entirely disappeared (Model 2). In regards to the specific diseases in the adjusted model, the IRR in the early-reproductive age group was significantly higher than that of late-reproductive women for viral intestinal infection (IRR = 2.24; 95% CI: 1.33–3.78), gastroenteritis and colitis (IRR = 2.20; 95% CI: 1.73–2.80), low back pain (IRR = 2.11; 95% CI: 1.77–2.52), hemorrhage in early pregnancy (IRR = 33.20; 95% CI: 4.32–246.4), nausea and vomiting (IRR = 2.32; 95% CI: 1.37–3.94), and abdominal and pelvic pain (IRR = 1.86; 95% CI: 1.16–2.98).

## Discussion

The results of this study show that women in early-reproductive age have an increased risk of initiating a SA spell compared to the late-reproductive age group, this contrasts with previous literature that demonstrates an increased risk of SA with older age [9,16]. A possible explanation could be found in the healthy worker effect, which describes how healthy workers remain in the workforce longer than those who are not, due to early termination or retirement. In this sense, it is possible that the late-reproductive group has a relatively higher proportion of healthier individuals than the early- or middle-reproductive groups, especially as the healthy worker effect is stronger in women [17], potentially leading to the lower incidence of SA observed in late-reproductive female workers.

In addition, the moderately increased risk of SA in early-reproductive aged women appears to be almost entirely explained by diagnostic groups and not occupational characteristics. This suggests that for SA, the type of diseases for which a woman needs to take an absence is more influential on the likelihood a woman starts a SA spell than occupational category, type of contract, or country of origin.

In regards to the higher risk of gastroenteritis, colitis, and other viral intestinal infections in early-reproductive women, few studies have described the age-dependent epidemiology of these diseases in adults. An English study found a slight decreasing incidence of *Campylobacter* infections from women aged 25–29 through 40–44 [23], though the study did not offer

**Table 2. Incidence rates of sickness absence in female workers by age group, according to diagnostic group, contract, occupational category, and country of origin, Catalonia 2012–2014.**

| | Early-Reproductive | | | | | | Middle-Reproductive | | | | | | Late-Reproductive | | | | | |
|---|---|---|---|---|---|---|---|---|---|---|---|---|---|---|---|---|---|---|
| | SA Spells | | Person-years at risk | IR* | 95% CI | MD | SA Spells | | Person-years at risk | IR* | 95% CI | MD | SA Spells | | Person-years at risk | IR* | 95% CI | MD |
| | (n) | (%) | | | | | (n) | (%) | | | | | (n) | (%) | | | | |
| **Diagnostic Group (ICD10)** | | | | | | | | | | | | | | | | | | |
| Infectious (A00-B99) | 2,195 | 13.6 | 35,784 | 6.25 | 5.99-6.51 | 2 | 1,611 | 11.1 | 40,022 | 4.09 | 3.89-4.29 | 3 | 838 | 8.5 | 28,419 | 3.00 | 2.80-3.20 | 3 |
| Neoplasms (C00-D49) | 170 | 1.1 | 35,784 | 0.48 | 0.41-0.55 | 15 | 371 | 2.6 | 40,022 | 0.94 | 0.85-1.04 | 19 | 339 | 3.4 | 28,419 | 1.21 | 1.08-1.34 | 24.5 |
| Hematological (D50-D89) | 36 | 0.2 | 35,784 | 0.10 | 0.07-0.14 | 19 | 49 | 0.3 | 40,022 | 0.12 | 0.09-0.16 | 16 | 25 | 0.3 | 28,419 | 0.09 | 0.05-0.12 | 20 |
| Metabolic (E00-E89) | 65 | 0.4 | 35,784 | 0.19 | 0.14-0.23 | 10 | 79 | 0.5 | 40,022 | 0.20 | 0.16-0.24 | 19 | 57 | 0.6 | 28,419 | 0.20 | 0.15-0.26 | 25 |
| Mental (F01-F99) | 1,062 | 6.6 | 35,784 | 3.02 | 2.84-3.21 | 18 | 1,132 | 7.8 | 40,022 | 2.87 | 2.71-3.04 | 17 | 851 | 8.6 | 28,419 | 3.04 | 2.84-3.25 | 22 |
| Nervous system (G00-G99) | 274 | 1.7 | 35,784 | 0.78 | 0.69-0.87 | 2 | 340 | 2.3 | 40,022 | 0.86 | 0.77-0.95 | 4 | 290 | 2.9 | 28,419 | 1.04 | 0.92-1.16 | 5 |
| Ocular (H00-H59) | 166 | 1.0 | 35,784 | 0.47 | 0.40-0.54 | 5 | 139 | 1.0 | 40,022 | 0.35 | 0.29-0.41 | 5 | 164 | 1.7 | 28,419 | 0.59 | 0.50-0.68 | 11 |
| Aural (H60-H95) | 171 | 1.1 | 35,784 | 0.49 | 0.41-0.56 | 4 | 241 | 1.7 | 40,022 | 0.61 | 0.53-0.69 | 5 | 174 | 1.8 | 28,419 | 0.62 | 0.53-0.72 | 5 |
| Cardiovascular (I00-I99) | 112 | 0.7 | 35,784 | 0.32 | 0.26-0.38 | 16.5 | 302 | 2.1 | 40,022 | 0.77 | 0.68-0.85 | 17 | 233 | 2.4 | 28,419 | 0.83 | 0.73-0.94 | 17 |
| Respiratory (J00-J99) | 3,360 | 20.9 | 35,784 | 9.57 | 9.25-9.89 | 3 | 3,078 | 21.1 | 40,022 | 7.81 | 7.53-8.09 | 4 | 2,014 | 20.4 | 28,419 | 7.21 | 6.90-7.53 | 5 |
| Digestive (K00-K95) | 837 | 5.2 | 35,784 | 2.38 | 2.22-2.55 | 3 | 736 | 5.1 | 40,022 | 1.87 | 1.73-2.00 | 4 | 556 | 5.6 | 28,419 | 1.99 | 1.83-2.16 | 5 |
| Skin (L00-L99) | 145 | 0.9 | 35,784 | 0.41 | 0.35-0.48 | 7 | 146 | 1.0 | 40,022 | 0.37 | 0.31-0.43 | 11 | 106 | 1.1 | 28,419 | 0.38 | 0.31-0.45 | 9 |
| Musculoskeletal (M00-M99) | 3,075 | 19.1 | 35,784 | 8.75 | 8.45-9.07 | 15 | 2,631 | 18.1 | 40,022 | 6.68 | 6.42-6.93 | 11 | 2,147 | 21.8 | 28,419 | 7.69 | 7.36-8.01 | 13 |
| Genitourinary (N00-N99) | 692 | 4.3 | 35,784 | 1.97 | 1.82-2.12 | 6 | 772 | 5.3 | 40,022 | 1.96 | 1.82-2.10 | 6 | 373 | 3.8 | 28,419 | 1.34 | 1.20-1.47 | 8 |
| Obstetric (O00-O9A) | 1,120 | 7.0 | 35,784 | 3.19 | 3.00-3.28 | 20 | 623 | 4.3 | 40,022 | 1.58 | 1.46-1.71 | 17 | 12 | 0.1 | 28,419 | 0.04 | 0.02-0.07 | 23 |
| Congenital (Q00-Q99) | 16 | 0.1 | 35,784 | 0.05 | 0.02-0.07 | 15 | 19 | 0.1 | 40,022 | 0.05 | 0.02-0.07 | 33.5 | 11 | 0.1 | 28,419 | 0.04 | 0.02-0.06 | 52 |
| Other (R00-R99) | 1,588 | 9.9 | 35,784 | 4.52 | 4.30-4.74 | 3 | 1,290 | 8.9 | 40,022 | 3.27 | 3.10-3.45 | 4 | 786 | 8.0 | 28,419 | 2.81 | 2.62-3.01 | 5 |
| Accidents (S00-T88) | 820 | 5.1 | 35,784 | 2.34 | 2.18-2.50 | 14 | 801 | 5.5 | 40,022 | 2.03 | 1.89-2.17 | 15 | 705 | 7.2 | 28,419 | 2.52 | 2.34-2.71 | 19 |
| External causes (V00-Y99) | 142 | 0.9 | 35,784 | 0.40 | 0.34-0.47 | 12 | 167 | 1.2 | 40,022 | 0.42 | 0.36-0.49 | 14 | 142 | 1.4 | 28,419 | 0.51 | 0.42-0.59 | 25 |
| Health services (Z00-Z99) | 47 | 0.3 | 35,784 | 0.13 | 0.10-0.17 | 14 | 47 | 0.3 | 40,022 | 0.12 | 0.09-0.15 | 14 | 29 | 0.3 | 28,419 | 0.10 | 0.07-0.14 | 23 |
| **Contract** | | | | | | | | | | | | | | | | | | |
| Permanent | 11,321 | 73.9 | 26,130 | 44.19 | 43.37-45.00 | 5 | 11,080 | 80.3 | 32,458 | 34.67 | 34.03-35.32 | 6 | 7,849 | 86.1 | 24,243 | 32.96 | 32.23-33.69 | 7 |
| Temporary | 3,994 | 26.1 | 8,046 | 50.64 | 49.08-52.22 | 4 | 2,726 | 19.7 | 6,376 | 43.53 | 41.90-45.17 | 5 | 1,272 | 13.9 | 3,565 | 36.32 | 34.33-38.32 | 7 |
| **Occupational category** | | | | | | | | | | | | | | | | | | |
| Skilled non-manual | 3,277 | 21.4 | 8,284 | 40.25 | 38.88-41.63 | 5 | 3,131 | 22.7 | 9,328 | 34.07 | 32.88-35.27 | 5 | 1,712 | 18.8 | 5,752 | 30.19 | 28.76-31.62 | 6 |
| Unskilled non-manual | 8,730 | 57.0 | 17,131 | 52.07 | 50.97-53.16 | 4 | 6,991 | 50.6 | 18,646 | 38.11 | 37.22-39.01 | 5 | 4,371 | 47.9 | 12,087 | 36.85 | 35.76-37.94 | 7 |
| Skilled manual | 2,088 | 13.6 | 4,770 | 44.76 | 42.84-46.68 | 6 | 2,107 | 15.3 | 5,361 | 40.02 | 38.31-41.72 | 6 | 1,546 | 16.9 | 4,124 | 38.29 | 36.38-40.20 | 8 |
| Unskilled manual | 1,220 | 8.0 | 2,563 | 48.69 | 45.96-51.42 | 5 | 1,577 | 11.4 | 3,800 | 42.34 | 40.25-44.43 | 7 | 1,492 | 16.4 | 4,126 | 37.00 | 35.12-38.88 | 9 |
| **Country of Origin** | | | | | | | | | | | | | | | | | | |
| Spain and EU-28 | 13,354 | 87.2 | 28,935 | 47.10 | 46.30-47.90 | 5 | 12,144 | 88.0 | 33,846 | 36.46 | 35.81-37.11 | 5 | 8,176 | 89.6 | 25,091 | 28.83 | 28.21-29.46 | 7 |
| Other | 1,719 | 11.2 | 6,074 | 28.68 | 27.33-30.04 | 6 | 1,498 | 10.9 | 5,606 | 27.06 | 25.69-28.43 | 7 | 832 | 9.1 | 3,017 | 27.98 | 26.08-29.88 | 7 |
| **Overall** | 15,315 | 100 | 35,784 | 43.62 | 42.92-44.31 | 5 | 13,806 | 100 | 40,022 | 35.03 | 34.45-35.62 | 5 | 9,121 | 100 | 28,419 | 32.65 | 32.01-33.35 | 7 |

IR: incidence rate, 95% CI: 95% confidence interval, MD: median duration of SA in days.

*per 100 person-years.

**Table 3. Incidence rates of sickness absence in female workers by age group, according to disease, Catalonia 2012–2014.**

| Disease (ICD10) | Early-Reproductive | | | | Middle-Reproductive | | | | Late-Reproductive | | | |
|---|---|---|---|---|---|---|---|---|---|---|---|---|
| | SA Spells | | IR* | 95% CI | SA Spells | | IR* | 95% CI | SA Spells | | IR* | 95% CI |
| | (n) | (%) | | | (n) | (%) | | | (n) | (%) | | |
| Infectious (A00-B99) | 2,195 | 13.6 | 6.25 | 5.99–6.51 | 1,611 | 11.1 | 4.09 | 3.89–4.29 | 838 | 8.5 | 3.00 | 2.80–3.20 |
| Viral Intestinal Infection (A08-A08.8) | 151 | 6.9 | 0.43 | 0.36–0.50 | 126 | 7.8 | 0.32 | 0.26–0.38 | 56 | 6.7 | 0.20 | 0.15–0.25 |
| Gastroenteritis & Colitis (A09) | 851 | 38.7 | 2.42 | 2.26–2.59 | 634 | 39.4 | 2.16 | 1.48–1.73 | 298 | 35.6 | 1.07 | 0.95–1.19 |
| Unlocalized Viral Infection (B34-B34.9) | 261 | 11.9 | 0.74 | 0.65–0.83 | 252 | 15.6 | 0.64 | 0.56–0.72 | 149 | 17.8 | 0.53 | 0.45–0.62 |
| Neoplasms (C00-D49) | 170 | 1.1 | 0.48 | 0.41–0.55 | 371 | 2.6 | 0.94 | 0.85–1.04 | 339 | 3.4 | 1.21 | 1.08–1.34 |
| Malignant Breast Cancer (C50-C50.929) | 8 | 4.7 | 0.02 | 0.01–0.04 | 25 | 6.7 | 0.06 | 0.03–0.09 | 39 | 11.5 | 0.14 | 0.09–0.18 |
| Lipoma (D17-D17.9) | 3 | 1.8 | 0.01 | <0.01–0.02 | 7 | 1.9 | 0.02 | <0.01–0.03 | 15 | 4.42 | 0.05 | 0.03–0.08 |
| Uterine Fibroid (D25-D25.9) | 28 | 16.5 | 0.08 | 0.05–0.11 | 71 | 19.1 | 0.18 | 0.14–0.22 | 37 | 10.9 | 0.13 | 0.09–0.18 |
| Benign Ovarian Cancer (D27-D27.9) | 11 | 6.47 | 0.03 | 0.01–0.05 | 19 | 5.12 | 0.05 | 0.02–0.07 | 7 | 2.1 | 0.03 | 0.01–0.04 |
| Musculoskeletal (M00-M99) | 3,075 | 19.1 | 8.75 | 8.45–9.07 | 2,631 | 18.1 | 6.68 | 6.42–6.93 | 2,147 | 21.8 | 7.69 | 7.36–8.01 |
| Neck Pain (M54.2) | 258 | 8.4 | 0.73 | 0.65–0.82 | 201 | 7.6 | 0.51 | 0.44–0.58 | 158 | 7.4 | 0.57 | 0.48–0.65 |
| Low Back Pain (M54.3-M54.5) | 1,027 | 33.4 | 2.92 | 2.75–3.10 | 740 | 28.1 | 1.88 | 1.74–2.01 | 409 | 19.0 | 1.46 | 1.32–1.61 |
| Obstetric (O00-O9A) | 1,120 | 7.0 | 3.19 | 3.00–3.28 | 623 | 4.3 | 1.58 | 1.46–1.71 | 12 | 0.1 | 0.04 | 0.02–0.07 |
| Retained Products of Conception (O02.0) | 46 | 4.1 | 0.13 | 0.09–0.17 | 34 | 5.5 | 0.09 | 0.06–0.12 | 0 | 0 | 0 | |
| Spontaneous Abortion (O03-O03.9) | 45 | 4.0 | 0.13 | 0.09–0.17 | 67 | 10.8 | 0.17 | 0.13–0.21 | 0 | 0 | 0 | |
| Early Pregnancy Hemorrhage (O20-O20.9) | 94 | 8.4 | 0.27 | 0.21–0.32 | 58 | 9.3 | 0.15 | 0.11–0.19 | 2 | 16.7 | 0.01 | <0.01–0.02 |
| Vomiting During Pregnancy (O21-O21.9) | 88 | 7.9 | 0.25 | 0.20–0.30 | 28 | 4.5 | 0.07 | 0.05–0.11 | 0 | 0 | 0 | |
| Antepartum Hemorrhage (O46-O46.9) | 43 | 3.8 | 0.12 | 0.09–0.16 | 31 | 5.0 | 0.08 | 0.04–0.10 | 0 | 0 | 0 | |
| Other (R00-R99) | 1,588 | 9.9 | 4.52 | 4.30–4.74 | 1,290 | 8.9 | 3.27 | 3.10–3.45 | 786 | 8.0 | 2.81 | 2.62–3.01 |
| Abdominal & Pelvic Pain (R10-R10.9) | 195 | 12.3 | 0.56 | 0.48–0.63 | 139 | 10.8 | 0.35 | 0.29–0.41 | 68 | 8.7 | 0.24 | 0.19–0.30 |
| Nausea & Vomiting (R11-R11.2) | 195 | 12.3 | 0.56 | 0.48–0.63 | 107 | 8.3 | 0.27 | 0.22–0.32 | 60 | 7.6 | 0.21 | 0.16–0.27 |
| Dizziness (R42) | 109 | 6.9 | 0.31 | 0.25–0.37 | 118 | 9.1 | 0.30 | 0.25–0.35 | 74 | 9.4 | 0.26 | 0.20–0.33 |
| Unspecified Fever (R50.9) | 107 | 6.7 | 0.30 | 0.25–0.36 | 77 | 6.0 | 0.20 | 0.15–0.24 | 55 | 7.0 | 0.20 | 0.14–0.25 |
| Headache (R51) | 50 | 3.1 | 0.14 | 0.10–0.18 | 56 | 4.3 | 0.14 | 0.10–0.18 | 26 | 3.3 | 0.09 | 0.06–0.13 |
| Fatigue (R53-R53.83) | 97 | 6.1 | 0.28 | 0.22–0.33 | 57 | 4.4 | 0.14 | 0.11–0.18 | 59 | 7.5 | 0.21 | 0.15–0.25 |

IR: incidence rate, 95% CI: 95% confidence interval.

* per 100 person-years.

explanations for this trend. Of note, abdominal and pelvic pain, and nausea and vomiting are both symptoms of intestinal infections. With all of these diagnoses presenting higher risks in early-reproductive women, it is difficult to determine to what degree there might have been confluence of symptom and cause of the disease in assigning diagnoses.

For low back pain, abdominal and pelvic pain, and nausea and vomiting, early-reproductive age also significantly increases the risk of starting SA. Higher incidences of these common health conditions during pregnancy should be expected in early-reproductive women, who have the highest birthrate. Obstetric diagnoses, including early pregnancy hemorrhage, are the most dramatic example of the pattern of higher incidence rates of SA in early-reproductive women.

The trend of decreasing incidence of SA due to obstetric diagnoses as age increases is in accordance with the trend of a decreasing birthrate with older age in this population. Interestingly, there are other diagnostic groups in which the incidence trend significantly decreases from the highest rates in early-reproductive to lowest in late-reproductive women. Some of those diagnosis groups contain pathologies, which could be linked not only to factors related to age itself, but to the different birthrates along the life course, such as musculoskeletal,

genitourinary, and other poorly-defined diseases [24]. Low back pain, nausea and vomiting, and pelvic girdle pain are some of the most common diseases during pregnancy, and contribute to much of the SA throughout this physiological state [24–27]. The significantly higher IR and IRR observed for early-reproductive and, to a lesser extent (IRR not significantly higher), middle-reproductive groups compared to the late-reproductive women in this cohort, suggests that reproductive age is an important predictor for SA episodes due to these symptoms, which are known to be frequent in pregnancy. Additionally, the high incidence of SA in early-reproductive age due to obstetric diagnoses and low back pain coupled with the increased risk observed in early-reproductive age women, potentially due to pregnancy, may partially explain the higher overall IRR in early-reproductive women compared to late-reproductive women.

Furthermore, Arcas *et al.* found that SA due to low back pain had a peak in duration in women aged 26–35, but not in men of the same age [28]. This reinforces the idea that SA episodes for low back pain in early- and middle-reproductive age women in this cohort could be related to higher birthrates, since this increase was only described in women and SA spells in pregnancy have longer durations than the average SA length [4,29]. In this study, young-reproductive women exhibited the longest duration of SA due to musculoskeletal diagnoses of the three age groups. However, no conclusions about pregnancy and the increased IRR for low back pain in younger women can be made from this study, as it could be attributable to other factors such as child care and other unpaid domestic work demands [22,24].

These results suggest that higher birthrates could explain the increased risk of initiating SA observed in early-reproductive-aged women. While this may sound intuitive, it may be indicative that workplace accommodations or social benefits for pregnant women are insufficient. In a cohort study with hospital employees in Catalonia, Villar and colleagues found, that pregnant women most frequently left employment before birth through an SA spell, rather than an available pregnancy occupational risk benefit or maternity leave [30]. Studies have demonstrated that job adjustment can decrease SA incidence [31], and general absences [32] in pregnant women who reported needing adjustment, which could help improve future employment opportunities for women after childbirth [33]. Future studies are needed to confirm the relationship between pregnancy and the increased risk of SA in younger working women, and to identify why women opt for SA over other benefits designed to protect working pregnant women. Additionally, subsequent research should investigate if the longer duration of SA episodes in late-reproductive women observed in this study is a contributing factor to the decrease in SA incidence with decreasing birthrates. However, this current study does not initially seem to support that relationship, as the trend of decreasing incidence of SA with lower birthrates is not mirrored by an increasing trend in SA duration. Rather an increase in SA duration is only observed in the late-reproductive group.

The main limitation of this study is the absence of data indicating how many women were pregnant, and number of pregnancies prior to and during the study period. However, we were able to approximate the effects of pregnancy by stratifying by age based on birthrates. While the age grouping may potentially lead to classification bias, the decreasing incidence rates of obstetric-related SA in older age groups suggests that the groups accurately capture birthrates, and further supports the interpretation of our findings. Our data did not include information on family structure, which is closely related to the number of children a woman decides to have, and the age in which she has them [34], and could affect how they cope with the conflict between domestic tasks, work balance, and career development. Furthermore, data on full-time or part-time employment was not available for this study. Those who are employed full-time may be expected to incur more SA, though type of contract provides similar information about the occupational context, and was able to be analyzed in the study. Yet, the available

**Table 4. Estimated incidence risk ratio (IRR) for overall SA and specific diseases among reproductive and late-reproductive age groups in female workers, Catalonia 2012–2014.**

| | Model 1 | | Model 2 | |
| --- | --- | --- | --- | --- |
| | cIRR | 95% CI | aIRR | 95% CI |
| **Overall SA* §** | | | | |
| Late-Reproductive | 1 | | 1 | |
| Middle-Reproductive | 0.99 | 0.97–1.02 | 1.00 | 0.97–1.03 |
| Early-Reproductive | 1.09 | 1.06–1.12 | 1.04 | 1.01–1.06 |
| **Specific SA Diseases** ‡** | | | | |
| INFECTIOUS | | | | |
| Viral Intestinal Infection | | | | |
| Late-Reproductive | 1 | | 1 | |
| Middle-Reproductive | 2.17 | 1.29–3.66 | 2.02 | 1.19–3.43 |
| Early-Reproductive | 2.51 | 1.50–4.20 | 2.24 | 1.33–3.78 |
| Gastroenteritis & Colitis | | | | |
| Late-Reproductive | 1 | | 1 | |
| Middle-Reproductive | 1.92 | 1.51–2.44 | 1.92 | 1.50–2.44 |
| Early-Reproductive | 2.28 | 1.80–2.89 | 2.20 | 1.73–2.80 |
| Unlocalized Viral Infection | | | | |
| Late-Reproductive | 1 | | 1 | |
| Middle-Reproductive | 1.02 | 0.73–1.41 | 1.01 | 0.73–1.41 |
| Early-Reproductive | 0.99 | 0.72–1.38 | 0.97 | 0.69–1.35 |
| NEOPLASMS | | | | |
| Malignant Breast Cancer | | | | |
| Late-Reproductive | 1 | | 1 | |
| Middle-Reproductive | 0.39 | 0.15–1.07 | 0.52 | 0.19–1.44 |
| Early-Reproductive | 0.14 | 0.04–0.46 | 0.19 | 0.06–0.64 |
| Lipoma | | | | |
| Late-Reproductive | 1 | | 1 | |
| Middle-Reproductive | 0.33 | 0.11–0.96 | 0.32 | 0.11–0.95 |
| Early-Reproductive | 0.07 | 0.01–0.51 | 0.06 | 0.01–0.51 |
| Uterine Fibroid | | | | |
| Late-Reproductive | 1 | | 1 | |
| Middle-Reproductive | 1.31 | 0.72–2.38 | 1.29 | 0.70–2.37 |
| Early-Reproductive | 0.61 | 0.31–1.19 | 0.61 | 0.40–1.22 |
| Benign Ovarian Cancer | | | | |
| Late-Reproductive | 1 | | 1 | |
| Middle-Reproductive | 2.66 | 0.86–8.25 | 2.72 | 0.87–8.50 |
| Early-Reproductive | 0.82 | 0.21–3.16 | 0.85 | 0.22–3.31 |
| MUSCULOSKELETAL | | | | |
| Neck Pain | | | | |
| Late-Reproductive | 1 | | 1 | |
| Middle-Reproductive | 0.79 | 0.55–1.14 | 0.86 | 0.59–1.24 |
| Early-Reproductive | 1.12 | 0.80–1.58 | 1.17 | 0.82–1.66 |
| Low Back Pain | | | | |
| Late-Reproductive | 1 | | 1 | |
| Middle-Reproductive | 1.46 | 1.22–1.75 | 1.47 | 1.22–1.77 |
| Early-Reproductive | 2.10 | 1.76–2.49 | 2.11 | 1.77–2.52 |
| OBSTETRIC | | | | |

*(Continued)*

**Table 4.** (Continued)

| | Model 1 | | Model 2 | |
|---|---|---|---|---|
| | cIRR | 95% CI | aIRR | 95% CI |
| Early Pregnancy Hemorrhage | | | | |
| Late-Reproductive | 1 | | 1 | |
| Middle-Reproductive | 18.82 | 2.51–141.4 | 18.33 | 2.43–138.2 |
| Early-Reproductive | 32.58 | 4.40–241.0 | 33.20 | 4.32–246.4 |
| OTHER | | | | |
| Abdominal & Pelvic Pain | | | | |
| Late-Reproductive | 1 | | 1 | |
| Middle-Reproductive | 1.52 | 0.95–2.42 | 1.61 | 1.00–2.59 |
| Early-Reproductive | 1.78 | 1.13–2.81 | 1.86 | 1.16–2.98 |
| Nausea & Vomiting | | | | |
| Late-Reproductive | 1 | | 1 | |
| Middle-Reproductive | 1.35 | 0.77–2.34 | 1.42 | 0.81–2.47 |
| Early-Reproductive | 2.23 | 1.32–3.77 | 2.32 | 1.37–3.94 |
| Dizziness | | | | |
| Late-Reproductive | 1 | | 1 | |
| Middle-Reproductive | 1.08 | 0.67–1.72 | 1.14 | 0.71–1.83 |
| Early-Reproductive | 0.81 | 0.50–1.33 | 0.88 | 0.53–1.45 |
| Unspecified Fever | | | | |
| Late-Reproductive | 1 | | 1 | |
| Middle-Reproductive | 0.76 | 0.45–1.28 | 0.71 | 0.42–1.21 |
| Early-Reproductive | 0.91 | 0.55–1.50 | 0.90 | 0.54–1.49 |
| Headache | | | | |
| Late-Reproductive | 1 | | 1 | |
| Middle-Reproductive | 2.23 | 0.98–5.06 | 2.23 | 0.98–5.08 |
| Early-Reproductive | 1.40 | 0.59–3.29 | 1.27 | 0.54–3.03 |
| Fatigue | | | | |
| Late-Reproductive | 1 | | 1 | |
| Middle-Reproductive | 0.66 | 0.35–1.23 | 0.65 | 0.35–1.22 |
| Early-Reproductive | 1.47 | 0.86–2.53 | 1.40 | 0.81–2.43 |

cIRR = crude incidence risk ratio; aIRR = adjusted incidence risk ratio; 95% CI = 95% confidence interval.

* Adjusted for diagnostic group, occupational category, type of contract, and country of origin

** Adjusted for occupational category, type of contract, and country of origin

§ Poisson regression

‡ Negative binomial regression.

data on socioeconomic and occupational characteristics of the women included in the analysis allow us to account for the possible influence of the occupational context.

An important strength of this study is that it is founded in the linkage of two administrative registries, allowing for a large sample size and the calculation of incidence rates based on exact person-years. This represents one of the few studies examining SA trends by medical diagnosis in women in Southern Europe [24,25], which is characterized by a relatively weak welfare state in comparison to other Northern European countries, where most research on SA in pregnancy has been performed [35]. Identifying the diagnoses that contribute most to the overall effect of increased risk of SA in young-reproductive women can help inform allocation of healthcare resources, mitigation strategies in the workplace, and policy aimed to guard

workers' health. Similarly, this is one of the few studies that attempts to explore the incidence of SA in pregnancy-related diseases.

In summary, these results suggest that the differences in SA between early- and late-reproductive-aged women can largely be explained by the higher incidences of infectious, obstetric, and musculoskeletal diseases in younger working women with higher birthrates. Moreover, the higher risk of SA for these diseases in early-reproductive women suggests that greater efforts should be made in the workplace to accommodate younger women with such symptoms, especially for pregnant women for whom these are common diseases, and sources of SA. Building in more flexibility to current maternity leave policies would give pregnant women more options in managing their health during pregnancy, and could reduce dependence on SA.

## Supporting information

**S1 Dataset. SA Catalan birthrates.**
(XLSX)

## Acknowledgments

We thank the Spanish National Social Security Institute and the Catalonian Institute for Medical Evaluations for transferring data from the Continuous Working Life Sample and sickness absence registers, respectively. We would like to thank Julio C. Hernando-Rodríguez for his help with the data set.

## Author Contributions

**Conceptualization:** Andrew N. March, Fernando G. Benavides, Laura Serra.

**Data curation:** Monica Ubalde-Lopez.

**Formal analysis:** Andrew N. March.

**Investigation:** Andrew N. March.

**Methodology:** Andrew N. March, Rocío Villar, Monica Ubalde-Lopez, Laura Serra.

**Project administration:** Fernando G. Benavides, Laura Serra.

**Supervision:** Rocío Villar, Monica Ubalde-Lopez, Fernando G. Benavides, Laura Serra.

**Writing – original draft:** Andrew N. March.

**Writing – review & editing:** Rocío Villar, Monica Ubalde-Lopez, Fernando G. Benavides, Laura Serra.

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
