## [Decision Letter · Decision Letter 0]

17 Apr 2020

PONE-D-20-06019

Do birthrates contribute to sickness absence differences in women? A cohort study in Catalonia, Spain, 2012-2014

PLOS ONE

Dear Mr. March,

Thank you for submitting your manuscript to PLOS ONE. After careful consideration, we feel that it has merit but does not fully meet PLOS ONE’s publication criteria as it currently stands. Therefore, we invite you to submit a revised version of the manuscript that addresses the points raised during the review process.

We would appreciate receiving your revised manuscript by Jun 01 2020 11:59PM. To enhance the reproducibility of your results, we recommend that if applicable you deposit your laboratory protocols in protocols.io, where a protocol can be assigned its own identifier (DOI) such that it can be cited independently in the future. For instructions see: http://journals.plos.org/plosone/s/submission-guidelines#loc-laboratory-protocols

We look forward to receiving your revised manuscript.

Kind regards,

Maria Christine Magnus, MPH

Academic Editor

PLOS ONE

2. In ethics statement in the manuscript and in the online submission form, please provide additional information about the patient records used in your retrospective study. Specifically, please ensure that you have discussed whether all data were fully anonymized before you accessed them and/or whether the IRB or ethics committee waived the requirement for informed consent. If patients provided informed written consent to have data from their medical records used in research, please include this information.

3. One of the noted authors is a group or consortium [Research Group on Statistics, Econometrics and Health (GRECS)]. In addition to naming the author group, please list the individual authors and affiliations within this group in the acknowledgments section of your manuscript. Please also indicate clearly a lead author for this group along with a contact email address.

Reviewers' comments:

Reviewer's Responses to Questions

**Comments to the Author**

1. Is the manuscript technically sound, and do the data support the conclusions?

Reviewer #1: No

Reviewer #2: Partly

2. Has the statistical analysis been performed appropriately and rigorously? 

Reviewer #1: No

Reviewer #2: No

3. Have the authors made all data underlying the findings in their manuscript fully available?

Reviewer #1: No

Reviewer #2: Yes

4. Is the manuscript presented in an intelligible fashion and written in standard English?

Reviewer #1: Yes

Reviewer #2: No

5. Review Comments to the Author

Reviewer #1: The aim of the study was to investigate differences in sickness absence rates between different age groups with different fertility rates.

However there are some important methodological problems and the study do not add much knowledge to the area.

Their dependent variabel is numbers of sickness absence spells given as IR (numbers of SA spells/100 employment years).

The independent variable is three age groups with different fertility rates.

I have two main concerns regarding the methodology:

1)

How do they count employment years. No information is given regarding censoring and do they take into account other reasons for absence ie maternity leave. What about the duration of SA spells? Could it be, that older women have longer spells and thus fewer numbers of spells?

Risktime in the different is not stated anywhere - not in the tables nor in the text.

2)

They have no knowledge of pregnancies among the women, thus the independent variable is only a proxy

The methods section is very short and not sufficient informative. There is no flow chart explaining how the final population was reached and it is sparsely described in the text. At what dates do the study start and end?

Regarding results:

There is no table 1 describing the different age groups with regard to contract types, occupations, country of origin, mean age, mean duration of SA spells, risk time, numbers of women etc.

Regarding conclusion

They conclude that results suggest that differences in sickness absence between early- and late reproductive groups can largely be explained by higher incidence of infectious, obstetric and musculosceletal disorders in younger women with (they write lower birthrates, I think they mean higher birth rates) higher birthrates.

First the overall IRR in the fully adjusted model is 1.04, which in this study with many

uncertainties, is not compelling.

Second the found difference in IR of diseases related to pregnancy is not surprising, but I am not convinced that it can explain the difference, if any, in overall SA.

Reviewer #2: General comments:

This study is focusing on a topic that is becoming increasingly important as female employment rates are growing in many countries. The study does provide some interesting findings.

However, I have some major concerns:

• A major limitation is that the analyses do not include information about working hours, or part time vs full time employment. Since the extent of part time employment is widespread among women, this might be an important confounder, which has received no attention at all.

• It is not clear how the study brings new knowledge to the research field.

• The study is generally marked by a lack of precision. For example, it is unclear whether all sickness absence spells are included, or whether spells are recorded after a certain number of days.

• The study lacks important information about the quality of the research data.

Specific comments:

Abstract:

26

Please clarify this aim. What is meant by "distribution of sickness absence trends"? Sickness absence differentials between age groups? Trends in sickness absence across age groups? It is not clear what the term “distribution” contributes with in this aim.

30

What is the percentage? (How many percent do the 47879 female employees constitute of the total population of female employees in the current age groups during this period of time?)

32

It seems a bit odd that the study excludes women younger than 25, when women as old as 54 are included. Please clarify. Also, the age groups are quite large, is it possible to make them smaller? For example 25-29 in one group, and 30-35 in the next?

34

Do these registries not include information about working hours, or the proportion of part time /full time employment among these women? This is particularly important when analyzing the impact of fertility on sickness absence among women, because longer working hours probably apply more often to younger women who do not have children.

Introduction:

56

What kind of legislation? EU-legislation? Legislation in Catalonia? It is not clear if legislation actually has been changed, or if there simply is a growing recognition of the need for doing so. How is this study related to the (need for) changing legislation?

70

Is this the kind of legislation that were mentioned in the first paragraph? In that case, the legislation should not be introduced until it is addressed more fully, that is, in the third paragraph.

77

Is confusing that the term "SA" refers to both "sickness absence" (being away from work) and "sickness absence benefit" (social insurance benefit). Please use different terms to distinguish between these.

84-85

What does it mean that spells are "reported"? Is that the same as being recorded in the registry that your study applies? What separates spells that are recognized by a National Health Service physician from spells that are not? Is the physician involved even the first few days when the worker gets no compensation, after 3 days when the employer covers the employment, or from day 17 when the National Social Security covers the pay, or at some other time? If spells are not recorded in the registry from the first sick day, but rather after a particular length of duration, please consider if this implies that the dependent variable in this study is truncated, and if so, if other regression models are more suitable than Poisson and negative binomial regression.

99

In the Abstract there seem to be one aim specified, but in line 99 there seem to be several aims. Please be consistent.

Please provide more information about any maternity leave scheme, the overall fertility rate, mean age for first-time pregnant women, and the level of employment among pregnant women in Catalonia.

Methods:

104

Please give more details about the quality of the data. What is the purpose of these registries? What is the level of reliability? Are the registries characterized by any weaknesses?

119

Does the dependent variable not account for the wide extent of part time employment among Spanish women? Full time employed women will probably have many more SA spells than part time employed women. This should be accounted for in the analyses. If it is not possible to account for this, the implications for the findings should be thoroughly discussed.

Also, it is unclear if it includes all possible sickness absence spells, or only spells recognized by the National Health Service physicians.

144

This seems a bit odd because "affiliate-years" is not a widely accepted term. Please use more common language.

152

Please explain why these types of regression (Poisson and the negative bionomial) were applied.

158

Please provide some more information about the ethics approval. Is this paper part of a larger study that was approved by the Committee? If so, what kind of study is that?

Results:

163-165

Please consider the possibility that age differentials in SA partly result from age specific patterns in part-time employment. Considering that female employment has increased in Spain recently, one might expect that younger women are more likely to longer working hours than older women, (particularly if the latter group more often have children). If so, one would expect younger women to have higher rates of sickness absence, simply because they work longer hours. This implies that working hours is an important possible confounder that should be included in the analyses. If that is not possible, the possible implications of this limitation should be properly discussed.

It is confusing that the results are characterized as “significant” in the text, as statistical significance is not easily detectable from the tables. For example, it is possible that overlapping confidence intervals yield significantly different point estimates (IRs). I would recommend an extra column with a p-value for the trend (early-middle-late).

Discussion:

327

If the study’s main contribution to the research field is that it provides empirical analyses from Southern Europe, this should be included and elaborated on in the introduction.

335

According to the above text, younger working women have higher, not lower birthrates (which is stated here).

6. PLOS authors have the option to publish the peer review history of their article (what does this mean?). If published, this will include your full peer review and any attached files.

Reviewer #1: No

Reviewer #2: No

---

## [Author Response · Author response to Decision Letter 0]

14 Jun 2020

A document containing response to reviewers has been attached to this submission.

---

## [Decision Letter · Decision Letter 1]

9 Jul 2020

PONE-D-20-06019R1

Do birthrates contribute to sickness absence differences in women? A cohort study in Catalonia, Spain, 2012-2014

PLOS ONE

Dear Dr. March,

Thank you for submitting your manuscript to PLOS ONE. After careful consideration, we feel that it has merit but does not fully meet PLOS ONE’s publication criteria as it currently stands. Therefore, we invite you to submit a revised version of the manuscript that addresses the points raised during the review process.

We look forward to receiving your revised manuscript.

Kind regards,

Maria Christine Magnus, MPH

Academic Editor

PLOS ONE

Reviewers' comments:

Reviewer's Responses to Questions

**Comments to the Author**

1. If the authors have adequately addressed your comments raised in a previous round of review and you feel that this manuscript is now acceptable for publication, you may indicate that here to bypass the “Comments to the Author” section, enter your conflict of interest statement in the “Confidential to Editor” section, and submit your "Accept" recommendation.

Reviewer #1: All comments have been addressed

2. Is the manuscript technically sound, and do the data support the conclusions?

Reviewer #1: Partly

3. Has the statistical analysis been performed appropriately and rigorously? 

Reviewer #1: Yes

4. Have the authors made all data underlying the findings in their manuscript fully available?

Reviewer #1: Yes

5. Is the manuscript presented in an intelligible fashion and written in standard English?

Reviewer #1: Yes

6. Review Comments to the Author

Reviewer #1: Thank you for the very fine response.

I do think the manuscript has improved significantly with the changes and the added table.

Still, I would say that the overall risk of SA is (only) MODESTLY increased in the young age group. (1.01, 95CI 1.01-1.06). The small increase could very well be due to share of parttime workers or duration of SA spells or other confounding factors. Thus I would prefer the wording: A slightly

increased/a modestly increased risk (l.308-309, l.366 and other places).

However, the findings show the differences between age groups regarding reasons for SA, and it indicates that adjustment of work or allocation of healthcare ressources to pregnant workers can be necessary if wanting to reduce SA in the young age group.

Extra comments:

I would recommend to update the references regarding occupational exposure and health effects (ref 8).

l.155 "excludes som categories of civil servants" (5%) - which categories? please give some examples if possible.

The median duration of SA is in days, months? (please state it in the table and the text).

7. PLOS authors have the option to publish the peer review history of their article (what does this mean?). If published, this will include your full peer review and any attached files.

Reviewer #1: No

---

## [Author Response · Author response to Decision Letter 1]

19 Jul 2020

Responses to reviewer comments are included in the attached files.

---

## [Editor Report · Decision Letter 2]

4 Aug 2020

Do birthrates contribute to sickness absence differences in women? A cohort study in Catalonia, Spain, 2012-2014

PONE-D-20-06019R2

Dear Dr. March,

We’re pleased to inform you that your manuscript has been judged scientifically suitable for publication and will be formally accepted for publication once it meets all outstanding technical requirements.

Kind regards,

David Desseauve, MD, MPH, PhD

Academic Editor

PLOS ONE
---

## [Editor Report · Acceptance letter]

17 Aug 2020

PONE-D-20-06019R2 

Do birthrates contribute to sickness absence differences in women? A cohort study in Catalonia, Spain, 2012-2014 

Dear Dr. March:

I'm pleased to inform you that your manuscript has been deemed suitable for publication in PLOS ONE. Congratulations! Your manuscript is now with our production department. 

Kind regards, 

on behalf of

Dr. David Desseauve 

Academic Editor

PLOS ONE